

# Symmetry-based reciprocity: evolutionary constraints on a proximate mechanism

Marco Campennì[1,2] and Gabriele Schino[3]

[1] Stockholm Resilience Centre, Stockholms Universitet, Stockholm, Sweden
[2] Department of Psychology, Arizona State University, Tempe, AZ, United States
[3] Istituto di Scienze e Tecnologie della Cognizione, Consiglio Nazionale delle Ricerche, Rome, Italy

## ABSTRACT

**Background.** While the evolution of reciprocal cooperation has attracted an enormous attention, the proximate mechanisms underlying the ability of animals to cooperate reciprocally are comparatively neglected. Symmetry-based reciprocity is a hypothetical proximate mechanism that has been suggested to be widespread among cognitively unsophisticated animals.

**Methods.** We developed two agent-based models of symmetry-based reciprocity (one relying on an arbitrary tag and the other on interindividual proximity) and tested their ability both to reproduce significant emergent features of cooperation in group living animals and to promote the evolution of cooperation.

**Results.** Populations formed by agents adopting symmetry-based reciprocity showed differentiated "social relationships" and a positive correlation between cooperation given and received: two common aspects of animal cooperation. However, when reproduction and selection across multiple generations were added to the models, agents adopting symmetry-based reciprocity were outcompeted by selfish agents that never cooperated.

**Discussion.** In order to evolve, hypothetical proximate mechanisms must be able to stand competition from alternative strategies. While the results of our simulations require confirmation using analytical methods, we provisionally suggest symmetry-based reciprocity is to be abandoned as a possible proximate mechanism underlying the ability of animals to reciprocate cooperative interactions.

# INTRODUCTION

A complete understanding of any biological phenomenon requires addressing four separate (but interacting) aspects: its ontogeny, phylogeny, proximate causation and ultimate function (*Mayr, 1961*; *Mayr, 1982*; *Tinbergen, 1963*). Nevertheless, it is often the case that only one of these aspects is emphasized, at the expense of the others. The interactions between different aspects are often similarly ignored (*Hofmann et al., 2014*; *Fawcett, Marshall & Higginson, 2015*).

A paradigmatic example is the study of reciprocal cooperation in animals. Given the obvious problem of explaining how cooperative behaviors (i.e., behaviors that benefit other individuals) could be favored by natural selection, the study of the ultimate function

Corresponding author
Gabriele Schino, g.schino@istc.cnr.it

and selective mechanisms underlying cooperative behaviors has prevailed in the literature, while other aspects have been systematically neglected. Thus, the role of reciprocity in the evolution of cooperation has been a topic for debate for more than 40 years (*Trivers, 1971*; *Trivers, 2006*) while the study of the proximate mechanisms supporting the ability to reciprocate cooperative interactions has only recently been addressed.

The first to propose a list of possible proximate mechanisms underlying reciprocity were *Brosnan & De Waal (2002)*. Building on previous work by *De Waal & Luttrell (1986)*, *De Waal & Luttrell (1988)* and *De Waal (2000)*, they proposed three hypothetical proximate mechanisms: symmetry-based, attitudinal and calculated reciprocity. The latter two mechanisms were later elaborated by *Schino & Aureli (2009)*, *Schino & Aureli (2010a)* and *Schino & Aureli (2010b)*.

Our focus in this paper is on the first of the mechanisms proposed by *Brosnan & De Waal (2002)*, symmetry-based reciprocity. This was conceived as the simplest and least cognitively demanding of the three and, as such, it was supposed to be widespread in the animal kingdom. Symmetry-based reciprocity is supposed to operate whenever the choice of the recipient of cooperation is based on symmetrical aspects of the relationships between individuals. Symmetrical aspects of the relationships could include similarity in age or dominance rank or mutual association. This mechanism does not involve any form of score keeping of cooperation received and is therefore cognitively extremely simple. Importantly, given that characteristics such as age similarity or interindividual proximity are by definition symmetrical, the resulting choice of partner is necessarily reciprocal, meaning that if individual A is a preferred partner of B, B will necessarily be among the preferred partners of A. Symmetry-based reciprocity was thus proposed to explain the positive correlation between cooperation given and received across pairs of individuals that is often observed in group-living animals (see *Schino & Aureli, 2010b*, for a review). Note that the term reciprocity is generally used to refer to any contingent cooperative investment that is based on the cooperative returns (*Carter, 2014*). As such, symmetry-based reciprocity would not fit the definition. Strictly speaking, in symmetry-based reciprocity what is "reciprocal" is the outcome of a process (symmetrical choices) and not a cause (contingency on received cooperation) of the process.

Reciprocity of cooperative interactions can in principle result from two different processes, first distinguished by *Bull & Rice (1991)*: partner fidelity (later called partner control by *Noë, 2006*) and partner choice. Partner control models conceive dyads of interacting individuals as conceptually isolated, so that the choice to behave cooperatively or not depends only on the past behavior of the partner. Partner choice models include a comparative component, so that individuals chose the partner to which they direct their cooperative behavior on the basis of a comparison of cooperation received (or anyhow available) from the different potential partners. Symmetry-based reciprocity is a hypothetical proximate mechanism that is clearly part of a partner choice process, since animals are supposed to make their decisions on the basis of a comparison of the available partners (animals are supposed to choose to cooperate with those most similar to themselves). Note, however, that no account of cooperation received is taken.

The acceptance of the concept of symmetry-based reciprocity relies on its generating behavior as seen in simple cooperative societies. Biological phenomena, however, require explanation in terms of all of the four interrelated aspects mentioned above. In particular, although proximate causation and ultimate function are logically separate issues, it is clear that any hypothesis addressing one must also be compatible with what is known about the other. The interactions between proximate causation and ultimate function have been most often instantiated in terms of the constraints that the former can impose on the latter (e.g., *Stevens & Hauser, 2004*, for reciprocity; *Holekamp, Swanson & Van Meter, 2013*, for behavioral flexibility; *Gould & Lewontin, 1979*, for a more general argument). It is our suggestion that the reverse constraint can apply in the case of symmetry-based reciprocity. In other words, we suggest (and test in this paper) that even if symmetry-based reciprocity provides a plausible mechanism explaining how animals behave, it cannot be accepted as a valid explanation because it is not a mechanism that can evolve, that is, natural selection will always eliminate from an evolving population those individuals that behave according to symmetry-based reciprocity.

Symmetry-based reciprocity can be considered as belonging to a set of models in which cooperation is directed to "similar" individuals. This is, however, a rather heterogeneous set that includes at least five different subgroups:

1. Models of kin-selected cooperation based on phenotype matching (*Rousset & Roze, 2007*). In these models, cues used to choose the preferred recipients of cooperation are genetically determined and identify kin (similarity depends on common descent). Cooperation is ultimately favored by kin selection (*Hamilton, 1964*).

2. Models of tag-based cooperation in which the cue (tag) used to identify the preferred recipients of cooperation is genetically determined and also encode cooperation (or cooperation is encoded by closely linked genes). This is the so called "green beard" effect, that is however vulnerable to invasion by mutants that carry the tag but do not cooperate (*Riolo, Cohen & Axelrod, 2001*; *Lehmann & Keller, 2006*; *Gardner & West, 2010*).

3. Models of tag-based cooperation in which the cue (tag) used to identify the preferred recipients of cooperation is genetically determined and does not encode cooperation. In these models, cooperation cannot generally evolve unless the population is highly structured, leading to preferential interactions between kin and, ultimately, kin-selected cooperation (*Hammond & Axelrod, 2006a*; *Hammond & Axelrod, 2006b*).

4. Models of tag-based cooperation in which the cue (tag) used to identify the preferred recipients of cooperation is cooperativeness itself. These are models of indirect reciprocity based on reputation or competitive altruism. They show cooperation can evolve, provided the tag used to identify preferred recipients is indeed informative of the recipient's behavior (*Roberts, 1998*; *Nowak & Sigmund, 2005*).

5. Models of tag-based cooperation in which the cue (tag) used to identify the preferred recipients of cooperation is not genetically determined. This is true symmetry-based reciprocity where tags can include characteristics such as age or dominance rank.

As already mentioned, we focus on symmetry-based reciprocity, that is, on the last of the above subgroups of models. While the first four subgroups above have attracted a great deal

of attention, symmetry-based reciprocity has been somewhat neglected by theoreticians. In this study, we tested the hypothesis (detailed above) that symmetry-based reciprocity cannot evolve by developing a set of agent-based models of symmetry-based reciprocation of cooperative interactions. First, we tested whether groups of agents adopting a strategy of symmetry-based reciprocation do reproduce features of the distribution of cooperative behavior observed among group living animals, i.e., if symmetry-based reciprocity can indeed result in the phenomenon it was originally conceived to explain. Second, we tested whether agents adopting symmetry-based reciprocation are evolutionarily successful against selfish agents that do not cooperate.

## MATERIALS & METHODS

Agent-based models were implemented using the NetLogo platform (NetLogo 5.0.5; *Wilensky, 1999*). Statistical analyses were conducted in R, version 2.14.2 (*R Development Core Team, 2012*). Social Network Analyses were conducted using Gephi 0.8.1 beta (*Bastian, Heymann & Jacomy, 2009*). A description of the models using the standard ODD (Overview, Design concepts, Details) protocol and the source code of all models are included in the Supplemental Information 1.

### "Single-generation" models

We developed these models to test whether two simple strategies of symmetry-based partner choice can reproduce significant emergent features of cooperation in group living animals. In a first model, partner choice was based on an observable arbitrary characteristic of the partners, and agents chose partners in relation to their similarity to themselves. We call this model "Tag Model" (file "Tag_SingleGen_Mdl.nlogo" in the Supplemental Information 1). This first model was not spatially explicit, i.e., agents were not set in space. In a second model agents were set in space and chose their partner in relation to their spatial proximity. We call this model "Proximity Model" (file "Prox_SingleGen_Mdl.nlogo" in the Supplemental Information 1).

Agents were created and equipped with a behavioral strategy, which differed in the two models (see below). In the Tag Model, each agent was also assigned a "tag", i.e., a random float number between 0 and 1. In the Proximity Model, each agent was randomly assigned an initial position in a 101 by 101 cells 2D toroidal space. Note that in both models agents had no memory of past interactions.

At each step of the simulation, all agents behaved cooperatively as explained below. First, an agent (the "actor") is randomly selected from the whole population of $N$ agents. Then, a subset of other agents (the "candidates") is randomly extracted among the remaining agents. In the Tag Model, the actor compares its own tag with the tags of the candidates, and directs its cooperative behavior towards the candidate whose tag is the most similar to its own (that is, the actor calculates the absolute differences between its own and the candidates' tags and chose the candidate with the smallest absolute difference). In the Proximity Model, first the actor moves following a simple random walk (i.e., selects a direction of travel randomly, and moves one unit length), then it calculates the distance between itself and the candidates. Finally, the actor directs its cooperative behavior towards

**Table 1** Parameters used to run the "single-generation" models.

| Parameter | Values |
| --- | --- |
| Population size ($N$ of agents) | 50 |
| Candidates for the interaction ($N$ of agents) | 2, 10, 25, 49 |
| Number of steps per simulation | 1,000 |
| Number of simulations (replicates) | 100 |

the closest candidate (if two or more candidates are equally close, the choice is random between them).

All agents in the population go through this sequence at each step of the simulation. The process is asynchronous and the order of agents is randomly chosen at each step of the simulation.

The number of candidates was varied systematically as summarized in Table 1. The output of each simulation was a sociometric matrix of the cooperation given by each agent to each other agent in the population. For each simulation, we calculated the within-subject linear regression between cooperation given and received to/by each other agent. We also calculated two common social network measures (centralization index and modularity) and produced figures representing the social networks of cooperation exchanged between agents.

### "Multi-generation" evolutionary models

We developed two evolutionary agent based variants of both the Tag Model and the Proximity Model in order to test whether a strategy of partner choice based on an arbitrary tag or on spatial proximity can promote the evolution of cooperation. Agents were created and equipped with a behavioral strategy (see below) and other properties (a nongenetically determined tag for the agents in the Tag Model and a position in space for the agents in the Proximity Model). Behavioral interactions had fitness costs and benefits, and the population composition varied generation after generation depending on the evolutionary success of the different strategies agents adopted.

#### "Two-strategy" models

In a first variant of the evolutionary models, agents were created that adopted one of two different behavioral strategies, choosing cooperative or selfish. Choosing cooperators behaved as described in the single-generation models (i.e., they chose their partner based on its tag in the Tag Model, and based on its spatial proximity in the Proximity Model). Selfish agents never cooperated, but could be the recipient of cooperative behavior by agents adopting the choosing cooperative strategy. At each step of the simulation, each agent behaved according to its own strategy (files "Tag_MultiGen_Mdl1.nlogo" and "Prox_ MultiGen_ Mdl1.nlogo" in the Supplemental Information 1).

Cooperation implied a cost for the actor and a benefit for the recipient. The fitness of each agent was calculated as the difference between the accumulated benefits received and costs incurred during a generation cycle. The selection process consisted in selecting the
**Table 2  Parameters used to run the evolutionary models.**

| Parameter | Values | |
|---|---|---|
| | "Two-strategy" model | "Continuous probability" model |
| Population size ($N$ of agents) | 50 | 50 |
| Candidates for the interaction ($N$ of agents) | 2, 10, 25, 49 | 2, 10, 25, 49 |
| Number of steps per generation | 1,000 | 1,000 |
| Number of generation per simulation | 50 | 200 |
| Number of simulations (replicates) | 30 | 30 |
| Benefit of receiving cooperation (fitness units) | 1.1, 2, 5, 50 | 1.1, 2, 5, 50 |
| Cost of cooperation (fitness units) | 1 | 1 |
| Number of strategies | 2 (choosing cooperative, selfish) | 1 (variable prob. of behaving cooperatively) |
| Initial strategy ratio or distribution of prob. of behaving cooperatively | 45/5, 25/25, 5/45 | $0.05 \pm 0.05$, $0.5 \pm 0.35$, $0.95 \pm 0.05$ |
| Mutation rate | 0.1 | 0.1 |
| Mutation effect | Switch strategy | $\pm 0.2$ |
| Proportion of agents selected for reproduction | 0.2 | 0.2 |

20% of agents with the lowest fitness values at the end of each generation and in removing them from the population. In order to keep population size stable, the 20% of agents with the highest fitness was made replicate themselves (see Table 2 for details about mutation rates). In the Tag Model, the tag of each agent was initialized at the beginning of each new generation. Note that this corresponds to a model with non overlapping generations in which 20% of the agents died without reproducing, 60% died and had one offspring (in the NetLogo code, they remained in the population, but had their tag initialized), and 20% died and had two offspring. Halving the intensity of selection did not change the results (data not shown). Note also that initializing the tag at each generation implies that the tag is not genetically determined. In the Proximity Model, each agent was assigned a new position in space at the beginning of each new generation. Again, this corresponds to a model with non overlapping generations with no spatial structuring and thus no preferential interaction between kin.

We varied the initial proportions of the two strategies, the benefit of receiving cooperation and the constraint on the cooperator's choice (i.e., the number of candidates for the receipt of cooperation). Cost of cooperation did not vary. Details about parameters used in these simulations are shown in Table 2.

### "Continuous probability" models

In a second variant of the evolutionary models, rather than having two discrete strategies (choosing cooperative or selfish), agents were characterized by an individual probability of behaving cooperatively (in the vocabulary of evolutionary game theory, these would be "mixed strategies" rather then "fixed strategies") (files "Tag_MultiGen_Mdl2.nlogo" and "Prox_MultiGen_Mdl2.nlogo" in the Supplemental Information 1). At each step of the simulation, each agent could behave as a choosing cooperator (as described above) with probability P or as a selfish agent with probability 1-P. We created agents and assigned

each of them an individual probability P of behaving as a choosing cooperator. Note that the probability of behaving cooperatively is a fixed characteristic of each individual agent and does not depend on the behavior of other agents. P values were assigned following a normal distribution whose mean and standard deviation were varied as described in Table 2. Other parameters of the simulations are also summarized in Table 2. At the end of each generation, selection and reproduction occurred as described for the previous models.

## RESULTS

### "Single-generation" models

These models evaluated if a population of cooperators adopting symmetry-based reciprocity can reproduce emergent features of cooperation in group living animals. In the Tag Model partner choice was based on an observable characteristic of the partners (the tag), and agents chose partners in relation to their similarity to themselves. In the Proximity Model agents were set in space and chose partners in relation to their spatial proximity.

An analysis of the behavior of agents in the Tag Model showed that they reproduced two features of cooperation in group living animals. First, they showed differentiated social relationships, that is, some pairs cooperated frequently, others less frequently, and others rarely if ever (Fig. 1A). Second, when correlations between cooperation given and received across pairs were calculated, significant positive relations emerged (Fig. 1B, and Table S2). When the constraint on the free choice of interactant was progressively decreased, results showed that social network differentiation increased, while the correlation between cooperation given and received first increased and then decreased (Figs. S1 and S2 and Tables S1 and S2). Also, a progressive formation of clusters of pairs appeared.

The analysis of the behavior of agents in the Proximity Model yielded similar results. Agents showed differentiated social relationships and a positive relation between cooperation given and received emerged (Fig. 1 and Table S2). Decreasing the constraint on the free choice of interactant had reduced effects in this model. Social network differentiation increased moderately, and only a slight reduction in the correlation between cooperation given and received was observed (Figs. S3 and S4 and Tables S1 and S2).

### "Multi-generation" evolutionary models

We developed these models in order to test the evolutionary success of symmetry-based reciprocity. In these models cooperation had fitness costs (for the actor) and benefits (for the recipient), agents had different behavioral options (cooperate or not) and reproduced at the end of each generation cycle. Selection operated in relation to the fitness (as derived from the accumulated costs and benefits) of each agent. We developed two evolutionary variants of both the Tag Model and the Proximity model.

In the "two-strategy" variant of the evolutionary Tag Model there were two kinds of agents adopting either a cooperative (symmetry-based) or a selfish strategy. Regardless of the parameters of the model, cooperative agents were always outcompeted by selfish agents. Populations initially formed by a majority of cooperative agents were always invaded by

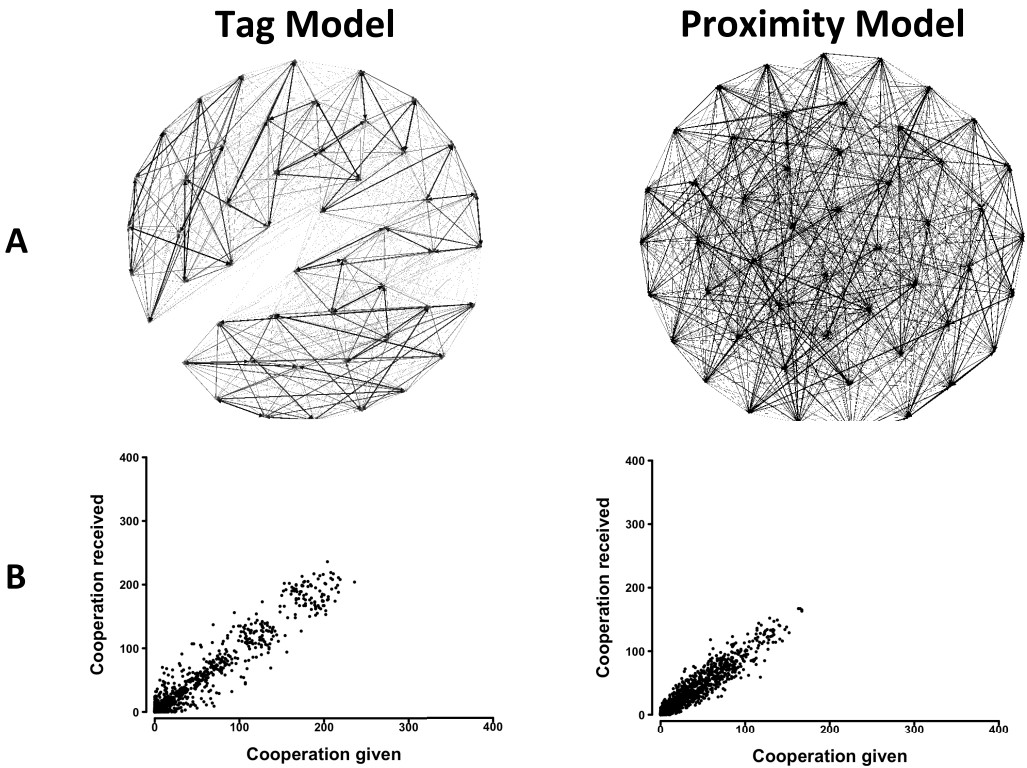

**Figure 1** **The distribution of cooperative interactions among agents in the Tag and Proximity Models.** (A) Networks of cooperation exchanged. (B) Cooperation given in relation to cooperation received. Representative distributions in simulations in which agents could make their choice between 10 randomly selected other agents. See the Supplemental Information 2 for the effects of varying additional parameters.

selfish agents, and populations initially formed by a majority of selfish agents were never invaded by cooperative agents (Fig. 2 and Figs. S5–S8).

The "two-strategy" variant of the evolutionary Proximity Model adopted the same two discrete strategies, cooperative and selfish. Similarly to what happened in the Tag Model, also in the Proximity Model cooperative agents were never successful (Fig. 3 and Figs. S9–S12 in the Supplementary Results).

In the "continuous probability" variant of the evolutionary Tag Model, rather than having two discrete strategies, agents were characterized by an individual probability of behaving (symmetry-based) cooperatively. The results confirmed the selective disadvantage of symmetry-based cooperation. Probability of behaving cooperatively decreased to zero along generations regardless of the parameters of the model and of the initial composition of the population (Fig. 4 and Figs. S13–S16).

Similarly, in the "continuous probability" variant of the evolutionary Proximity Model symmetry-based cooperation was never able to survive in the population (Fig. 5 and Figs. S17–S20).
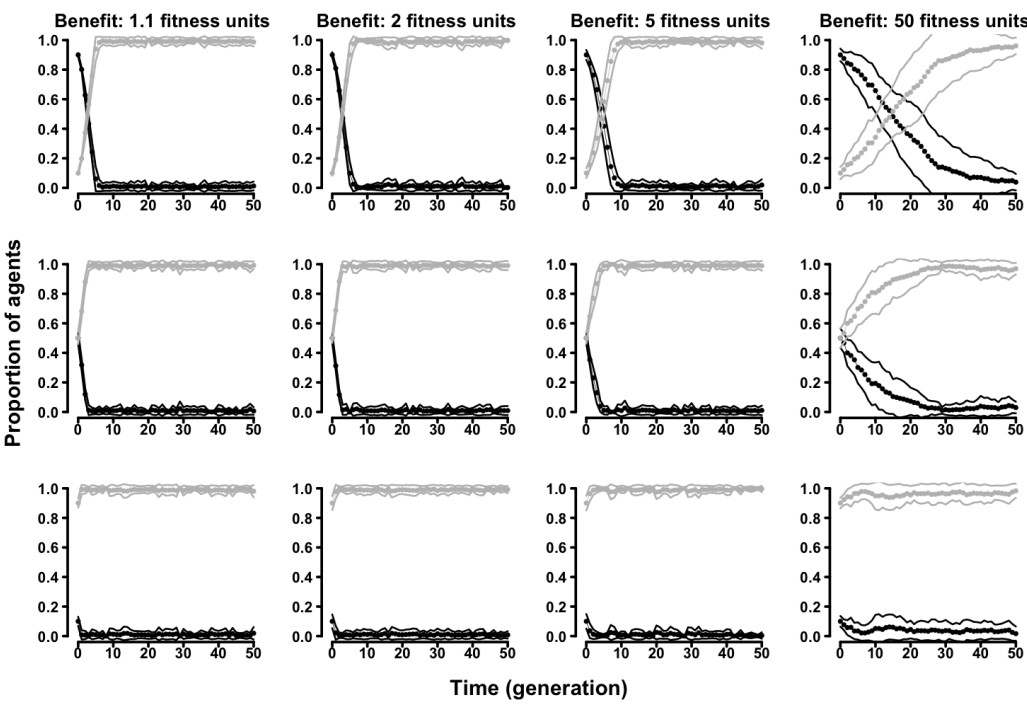

**Figure 2** **The evolution of symmetry-based reciprocity in the "two-strategy" Tag Model.** Symmetry-based cooperators: black dots; selfish: grey dots. Populations varied in relation to their initial composition and in the fitness benefits of receiving cooperation. Cost of cooperation for the actor was always one fitness unit. Choosing cooperators could make their choice among 10 randomly selected other agents. See the Supplemental Information 2 for the effects of varying additional parameters.

## DISCUSSION

Our first set of results showed that an agent-based model of symmetry-based reciprocity reproduces aspects of cooperation in group living animals. Our second and more important set of results, however, showed that when reproduction and selection are added to the model, selfish agents always outcompete agents adopting symmetry-based reciprocity. Although symmetry-based reciprocity appears as a plausible proximate mechanism underlying reciprocal exchanges, it is not evolutionary viable and thus seems to fail as a complete biological explanation of reciprocal cooperation. Indeed, the correspondence between our first set of results and actual cooperative behavior shown by group-living animals appears to be purely phenomenological, as symmetry-based reciprocity cannot evolve and cannot thus underlie the behavior of real animals.

Before further discussing the implications of these findings, we have to acknowledge their limitations. First, we modeled altruistic behaviors that have a net cost to the actor. It remains to be tested whether symmetry-based reciprocity might play a role in the exchange of mutualistic behaviors, that imply benefits for both the actor and the receiver. Second, simulation studies such as ours cannot in principle be used to demonstrate an impossibility, since they cannot of course cover all theoretically possible combinations of parameters. Therefore, while the results of our study do suggest symmetry-based reciprocity cannot

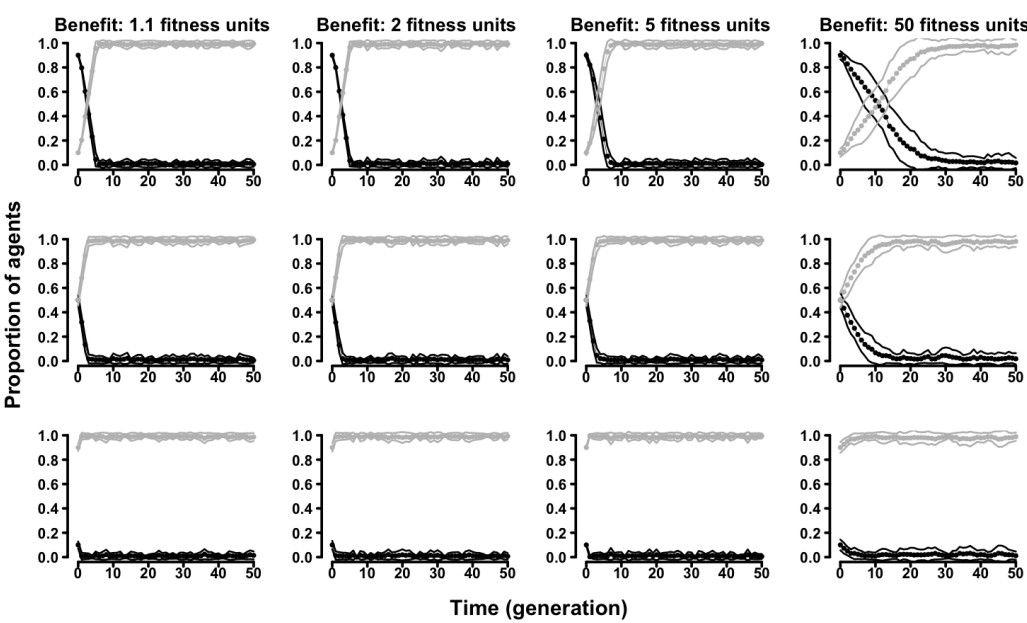

**Figure 3  The evolution of symmetry-based reciprocity in the "two-strategy" Proximity Model.**
Symmetry-based cooperators: black dots; selfish: grey dots. Populations varied in relation to their initial composition and in the fitness benefits of receiving cooperation. Cost of cooperation for the actor was always one fitness unit. Choosing cooperators could make their choice among 10 randomly selected other agents. See the Supplemental Information 2 for the effects of varying additional parameters.

evolve, a conclusive proof would require an analytical demonstration. Our conclusions have therefore to be considered provisional and awaiting analytical confirmation.

Our conclusions are coherent with those from the literature on indirect reciprocity, that show how the evolution of cooperation is contingent on the availability of information about the behavior of the potential recipients of cooperation (*Nowak & Sigmund, 1998*; *Rand & Nowak, 2013*). Since in our study the tags used to judge similarity (or proximity) do not encode any information about cooperativeness (either directed to self or to others), selfish cheaters were not avoided and prevailed during evolution. Recently, *Rauwolf, Mitchell & Bryson (2015)* showed that in a model of indirect reciprocity, the tendency to interact with individuals sharing similar beliefs (that may be considered analogous to an arbitrary tag) facilitated cooperation. While this may suggest a role for symmetry in models of indirect reciprocity, its relevance would be limited to humans, as indirect reciprocity seems to be rare in nonhuman animals, possibly because the absence of language constraints the efficient spread of information. Symmetry-based reciprocity, in contrast, has been explicitly proposed (and is typically invoked) to explain reciprocal cooperation in cognitively limited animals (*De Waal & Suchak, 2010*), where indirect reciprocity appears to be absent.

As already noted, biological phenomena require explanation in terms of four logically separate but interacting aspects. Confusion about these different levels of explanation has most often resulted in assuming unrealistically complex proximate mechanisms for reciprocal cooperation, i.e., in assuming that the delayed return benefits that characterize evolutionary explanations of reciprocity also play a motivational (proximate) role, and thus that reciprocity requires some understanding of future events (*De Waal, 2008*). Symmetry-based

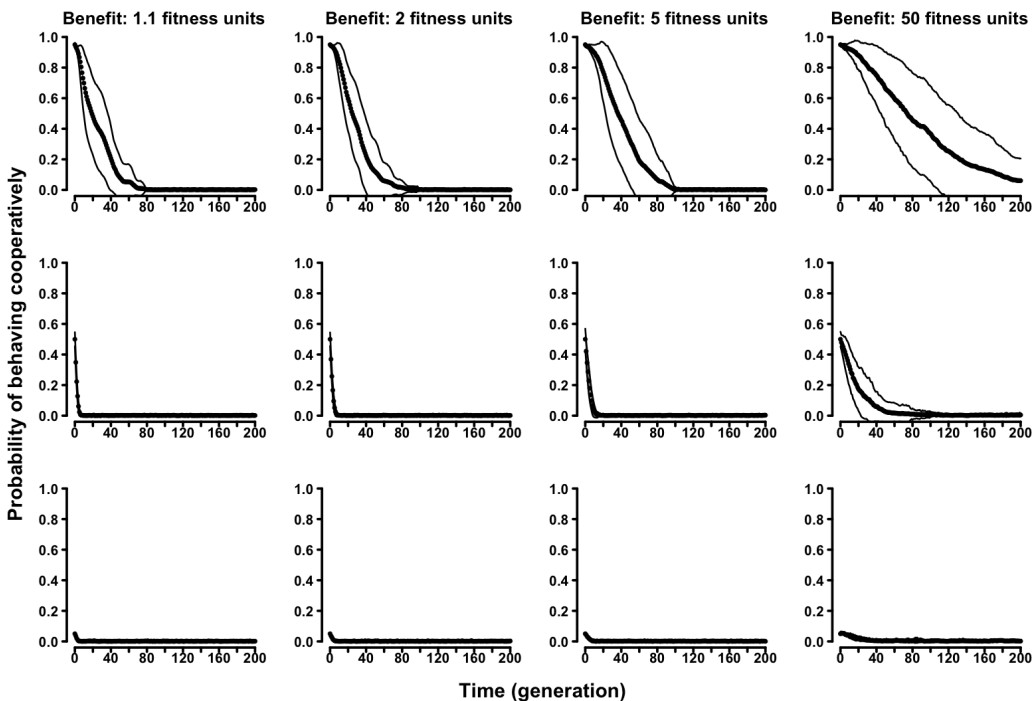

**Figure 4 The evolution of symmetry-based reciprocity in the "continuous probability" Tag Model.** Populations varied in relation to their initial composition and in the fitness benefits of receiving cooperation. Cost of cooperation for the actor was always one fitness unit. Agents could make their choice among 10 randomly selected other agents. See the Supplemental Information 2 for the effects of varying additional parameters.

reciprocity resulted to have the opposite shortcoming. It provides a plausible proximate mechanisms for reciprocal cooperation that is not, however, evolutionarily robust.

Doubts about the evolutionary viability of symmetry-based reciprocity had already been expressed (*Silk, 2005*; *Schino & Aureli, 2010b*). Nevertheless, in the absence of an explicit evolutionary test, symmetry-based reciprocity persisted in the literature and is indeed often considered as a sort of baseline mechanisms that is to be invoked whenever it is not possible to demonstrate more complex mechanisms such as calculated reciprocity (*De Waal & Luttrell, 1986*; *Jaeggi, Stevens & Van Schaik, 2010*; *De Waal & Suchak, 2010*). Decision rules based on interindividual proximity are similarly invoked as a simple/baseline mechanism from which reciprocal cooperation may derive as a byproduct. The need to exclude any effect of interindividual proximity in order to demonstrate "true reciprocity" is thus emphasized (*Schino, Polizzi di Sorrentino & Tiddi, 2007*; *Balasubramaniam et al., 2011*; *Carne, Wiper & Semple, 2011*). Having shown its evolutionary weakness, we suggest symmetry-based reciprocity (based on either an arbitrary tag or interindividual proximity) is to be abandoned as a proximate explanation for the occurrence of reciprocity in cognitively unsophisticated animals. Reciprocal cooperation among group living animals that form stable social relationships is most likely supported by emotionally based reciprocity (*Schino & Aureli, 2009*; *Schino & Aureli, 2010b*), while reciprocity in the absence of stable social relationships (*Sella, 1985*; *Petersen, 1995*) does require a simpler mechanisms.

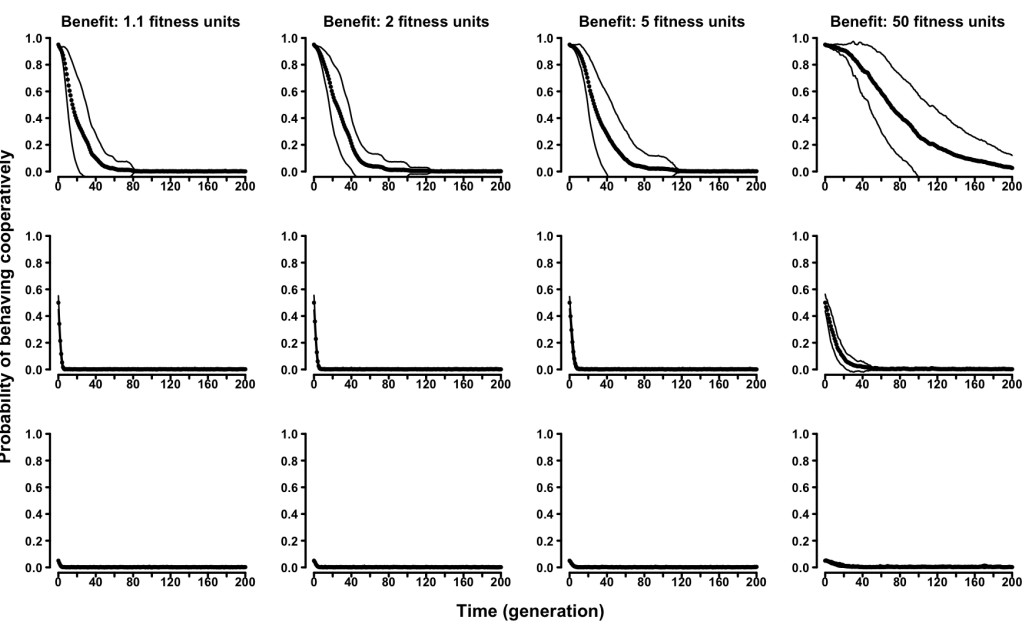

**Figure 5** **The evolution of symmetry-based reciprocity in the "continuous probability" Proximity Model.** Populations varied in relation to their initial composition and in the fitness benefits of receiving cooperation. Cost of cooperation for the actor was always one fitness unit. Agents could make their choice among 10 randomly selected other agents. See the Supplemental Information 2 for the effects of varying additional parameters.

Such simpler mechanism, however, cannot be symmetry-based reciprocity given its inability to survive to competition from alternative, selfish strategies.

It should be noted here that reciprocal cooperation based on stable social bonds has been sometimes misconceived as an example of symmetry-based reciprocity (*De Waal & Suchak, 2010*). This is incorrect, as the defining characteristic of symmetry-based reciprocity is the absence of any form of bookkeeping of cooperation received. In contrast, emotionally based reciprocity assumes the receipt of cooperative interactions triggers a partner-specific emotional response that translates into a social bond and acts as an emotionally based bookkeeping system of cooperation received. As such, emotionally based reciprocity succeeds as a proximate mechanism for reciprocal cooperation and is also evolutionarily viable (*Campennì & Schino, 2014*; *Evers et al., 2014*; *Evers et al., 2015*; *Evers et al., 2016*; *Puga-Gonzalez, Hoscheid & Hemelrijk, 2015*).

The ingroup bias (ethnocentric cooperation) frequently reported in the human literature might be considered as similar to symmetry-based reciprocity. However, human ethnocentric cooperation seems to derive from the need of tight and cooperative within-group relationships in the face of strong inter-group competition (*Bowles & Gintis, 2003*; *Bowles & Gintis, 2011*) and is therefore different from symmetry-based cooperation as is supposed to operate in cognitively limited animals. Other models of tag-based (or ethnocentric) cooperation have shown that the evolutionary success of tag-based cooperation requires spatially structured populations with high viscosity, so that agents cooperating with other similarly tagged agents often interact with their own offspring

(*Hammond & Axelrod, 2006b*). Eventually, it is kin selection that can insure the success of tag-based cooperators. In the absence of population viscosity and preferential interaction with kin, tag-based cooperation cannot evolve, coherently with our own results (*Axelrod, Ammond & Grafen, 2004*; *Hammond & Axelrod, 2006a*). It should be noted that in these models the tag used to guide decisions about cooperation is genetically determined, and thus coevolves with cooperation. In our model, on the contrary, the tag is not genetically determined and cannot evolve. It should also be noted that these previous models of tag-based cooperation were always partner control models in which cooperative interactions were included as one-shot prisoner dilemmas. In contrast, our models are partner choice models in which obligate cooperators chose the recipient of their cooperation on the basis of some defined rule (*Noë, 2001*; *Campennì & Schino, 2014*). Our preference for a partner choice model with a non genetically determined tag derived from the need to model symmetry-based reciprocity, particularly as acting in group living animals.

Previous agent-based models of cooperative exchanges in group living animals had emphasized the role of interindividual proximity in producing patterns of group distribution of cooperative exchanges that reproduce "apparent" reciprocity (*Puga-Gonzalez, Hildenbrandt & Hemelrijk, 2009*; *Hemelrijk & Puga-Gonzalez, 2012*). Our results confirmed that a spatially structured model in which decisions about cooperation are made according to simple interindividual proximity can generate patterns of cooperative exchanges that are similar to those observed in group living animals. Our models, however, went one step further by demonstrating that such simple decision rule is not evolutionarily stable.

In conclusion, the results of our study highlight how hypotheses about the proximate mechanisms underlying behavior cannot leave aside considerations about their evolvability. The integration of tests of proximate determinants and of ultimate functions provides a more compelling test of biological hypotheses and should be recommended whenever possible (*Akçay et al., 2009*). To the extent that the results of our simulations can be generalized (and awaiting their analytical confirmation), symmetry-based reciprocity seems to fail such a double test, and should therefore be abandoned as a hypothetical proximate mechanism supporting animal cooperation. It is left to the ingenuity of students of animal behavior to hypothesize a new biologically plausible proximate mechanism that could support reciprocity in cognitively unsophisticated animals that do not form stable social relationships (*Schino & Aureli, 2010b*).

## ACKNOWLEDGEMENTS

We thank Federico Cecconi for helpful discussions.

### Funding

We thank the John Templeton Foundation for funding the Human Generosity Project. The funders had no role in study design, data collection and analysis, decision to publish, or preparation of the manuscript.

### Grant Disclosures

The following grant information was disclosed by the authors:

John Temple Foundation.

Human Generosity Project.

### Competing Interests

The authors declare there are no competing interests.

### Author Contributions

- Marco Campennì conceived and designed the experiments, performed the experiments, analyzed the data, wrote the paper, prepared figures and/or tables, reviewed drafts of the paper.
- Gabriele Schino conceived and designed the experiments, wrote the paper, reviewed drafts of the paper.

### Data Availability

Source code for all the models can be found in the Supplemental Information.

### Supplemental Information

Supplemental information for this article can be found online at http://dx.doi.org/10.7717/peerj.1812#supplemental-information.

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
