# Peer review of "Symmetry-based reciprocity: evolutionary constraints on a proximate mechanism"

_PeerJ, doi:10.7717/peerj.1812_

## Round 0.1 · original submission · Major Revisions

Dear Marco Camepenini and Gabriele Schino,

Your article has now been reviewed by three independent reviewers. All the reviewers agree that your methods and results are sound. The reviewers’ concerns centered around your conceptual framework and review of the literature, and around the novelty of your results.

My reading of the manuscript and of the reviews suggests that you may be able to address reviewer concerns by thinking more carefully about how to cite and integrate past literature on cooperation, and about the organization of the Introduction vis-à-vis the relationship between symmetry-based reciprocity and other hypotheses in the literature. The reviewers pointed to several gaps in your discussions of the literature in this regard.. Some specific suggestions include the following. (1) Avoid being repetitive in stating what your goals are for this manuscript (for instances, lines 104-106 are similar to lines 136-138). (2) Outline a more clear and coherent description of how symmetry based reciprocity fits in with other hypotheses about cooperation; your current description is not quite thorough enough (see reviewer comments) and is poorly organized.

It also seems that two of the three reviewers raised major concerns about whether your result is sufficiently novel to warrant publication. In revising, please carefully address these concerns, and particularly clarify whether and how your result is different from, and links to, Nowak and Sigmund (1998), Rauwolf et al. (2015), and Nakamura and Masuda (2012)?

Reviewer 1 ·

Basic reporting

Generally, the work is clear and well-reported. The background is sufficient, but I do have a few concerns regarding how this work fits into the literature on ingroup biasing and indirect reciprocity. Please see the General Comments section for a more detailed explanation.

Also, a few minor corrections:

Line 156-157: Insert a period after the sentence: Agent-based models were implemented using the NetLogo platform (NetLogo 5.0.5;Wilensky, 1999)

Line 317-318: Fix the sentence: Symmetry-based reciprocity resulted to have a different, opposite shortcoming.

Experimental design

This work is scientifically sound. The simulations test the presented hypothesis and parameters are tested over a wide range of values, validating the generalizability of the result. I have no concerns regarding the experimental design.

Validity of the findings

I'm concerned with the novelty of this experiment. However, I believe grounding this work in a more in-depth discussion of ingroup biasing and indirect reciprocity may be sufficient to clarify how this work adds value to the literature. Please see the General Comments section for an in-depth discussion of my concerns.

Additional comments

Summary of the work:

This work employs an agent-based model to study the evolutionary viability of symmetry-based reciprocity. Symmetry-based reciprocity is defined as the tendency to cooperate with another based on a shared trait. In this work, two traits are analyzed --- an arbitrary tag and proximity. Simulations are run over one generation, then over multiple generations to diagnose whether symmetry-based reciprocity a.) generates behaviour reminiscent of empirical studies, and b.) is a viable evolutionary strategy.

The work uncovers that, while symmetry-based reciprocity generates behaviour matching empirical work, the strategy is not adaptive. As such, the authors suggest that symmetry-based reciprocity be abandoned as a proximate mechanism enabling the evolution of cooperation.

Concerns:

This work is scientifically sound. The simulations test the presented hypothesis and parameters are tested over a wide range of values, validating the generalizability of the result. However, a few concerns remain.

First, the authors claim that symmetry-based reciprocity engenders cooperative behaviour similar to that found in the experimental literature. Whilst it is important to test whether one’s model matches empirical data, it is not prima facie a result just because modelled behaviour looks similar to empirical data. For instance, the authors claim that their model generates variance in individual cooperation, similar to experimental studies. However, an infinite number of models can accomplish this. This important question is whether such a mechanism is adaptive. Since, in the second part of the work the authors demonstrate that symmetry-based reciprocity is not adaptive, it seems strange to argue that it is a finding that symmetry-based reciprocity causes behaviour reminiscent of empirical data. If symmetry-based reciprocity is not evolvable (second part of the work), then any similarity between this model’s behaviour and cooperation in the animal kingdom is a coincidence. The authors have proven that symmetry-based reciprocity is not evolvable, therefore it can’t be the cause of actual cooperative behaviour, even if it looks the same, because actual cooperative behaviour evolved.

Perhaps, however, this idea can be reframed. Is symmetry-based reciprocity a red-herring because it generates behaviour we’d expect in a cooperative society, even if it isn’t evolvable?

Second, and more importantly, I’m not sure the author’s second finding (symmetry-based reciprocity is not adaptive) is novel. Symmetry-based reciprocity is attempting to grapple with a well-known problem in the evolution of cooperation. In highly mobile societies (such as in human societies), direct reciprocity and kin-selection fail to create cooperation, and yet, we see large levels of cooperation in human societies.

The indirect reciprocity literature has found a solution to this problem. In their seminal work, Nowak and Sigmund (1998) showed that, in a well mixed population where agents do not possess information about others, cooperation cannot evolve. Information about another’s strategy is required for the evolution of cooperation in mobile societies. Their work ushered in fifteen years of attempting to analyze what signals are sufficient for cooperation to evolve.

The present work analyzes whether symmetry-based reciprocity can evolve. To do this, each agent is randomly tagged with an arbitrary marker or a location. However, since the marker is not correlated to behaviour, it gives no information about an agent. As such, I’m not sure there is a difference from Nowak and Sigmund’s finding.

Nowak and Sigmund demonstrated that, in a population without any information, cooperation cannot evolve. Isn’t the present work the same as Nowak and Sigmund’s, except that agent’s bias who plays with whom in an arbitrary fashion? It is well-known that cooperation cannot evolve in mobile populations unless defectors can be weeded from the population. Arbitrary markers do not attempt to solve this problem.

Please explain how your work differentiates itself from Nowak and Sigmund? Further, perhaps it would be useful to add a paragraph discussing how the present work is linked (or not) to the indirect reciprocity literature.

Links to the ingroup literature

Given that this work demonstrates that proximity is not sufficient to evolve cooperation, I think it would be beneficial to briefly explain how it fits into the literature on ingroup bias. At first glance, ingroup biasing seems to be symmetry-based reciprocity in disguise. For instance, Nakamura and Masuda, (2012) show that population structure (i.e. proximity) augments the evolution of cooperation. Can the present work help clarify why symmetry-based reciprocity in the guise of ingroup biasing is pervasively witnessed if it is not adaptive?

Rauwolf, et. al. (2015) mixes symmetry-based reciprocity with indirect reciprocity. In their work, symmetry is based on beliefs of others --- value homophily. Two agents who share an opinion of a third party have a higher chance of interacting. Rauwolf et. al. found that cooperation evolves if agents prioritize maintaining symmetry, even at the cost of spreading false information. Could symmetry-based reciprocity play a large part in the evolution of cooperation, even if it is not a sufficient condition? How you would explain explain these results in relation to the present work?

Summary:

The manuscript is technically sound and the simulations thoroughly analyzed. I have concerns regarding the novelty of the results. However, adding a more detailed discussion of how this work relates to the findings of indirect reciprocity and ingroup biasing should be sufficient to ground the findings, clarifying how the work augments the field.

References:

Nakamura, M., Masuda, N., 2012. Groupwise information sharing promotes ingroup favoritism in indirect reciprocity. BMC Evolutionary Biology 12 (1), 213.

Nowak, M. A., Sigmund, K., 11 June 1998. Evolution of indirect reciprocity by image scoring. Nature 393, 573–577.

Rauwolf, P., Mitchell, D. and Bryson, J. (2015) Value homophily benefits cooperation but motivates employing incorrect social information. Journal of Theoretical Biology, 367. 246-261

Reviewer 2 ·

Basic reporting

The manuscript reports results from an agent based modeling exercise designed to test predictions derived from the symmetry based reciprocity hypothesis. The hypothesis states that the evolution and maintenance of cooperation may result from altruistic (or cooperative) individuals preferentially directing their behavior at recipients that are similar to them in a trait or state that is not genetically fixed. The authors built models to test whether simple rules of partner choice based on self-similarity may produce empirical patterns of social structure and cooperation and whether individuals adopting these strategies can invade or survive invasion by purely selfish non-cooperative individuals. Results from these latter, evolutionary models imply that the implemented forms of symmetry based reciprocity cannot evolve. The writing is clear and concise and the motivation for the study well derived. The methods section provides all information necessary to replicate the study. The supplementary material is important for a full comprehension of results.

Experimental design

I only have a few minor comments concerning the formal aspects of the manuscript:

L 260 The wording is misleading here: decreasing a constraint implies more freedom, translating into more, not fewer partners in this model.

Please specify how the nearest neighbor is picked if two or more individuals are equally close to the actor (which must occur on a relatively small 101x101 grid with 50 individuals.

Why is it that axes in the scatter plots do not intersect?

The way S2 is plotted suggest that number of cooperative events decreases with increasing candidate pools. I understand that it really is the number of partners that changes and that fewer and fewer recipients concentrate all the cooperative events upon themselves. I suggest to either use bubble plots or to provide additional detail in the figure legend to avoid misinterpretation.

Validity of the findings

The way cooperation is modelled here suggests that single acts are truly altruistic whereas more and more empirical studies come to conclude that many cases of cooperation are best viewed as mutualistic exchanges with unavoidable benefits for both partners. It seems possible, therefore, that the strategies modelled here are relevant only for a fraction of the phenomena observed in social animals. The authors may wish to add a note on the scope of their analysis to the discussion.

I am concerned a bit about the strength of selection implemented in the models. Allowing only the 20% fittest individuals to reproduce and systematically culling the bottom 20% makes for a very high pace of evolution. This makes me wonder whether it is this strong selection that hampers the evolution of cooperation. Neutral models produce strong correlations between giving and receiving and strong population structure. If selection would be weaker emergence of this structure may be hampered only a little which then could promote the invasion of cooperators or save them from being invaded.

The assumption of random tag assigment is rather strong. The tags mentioned are dominance rank, age and spatial position all of which are not changing randomly. Please comment on the role of this assumption for outcome of the modeling in the discussion..

Additional comments

The paper is an important advancement over the evolutionarily neutral models on the emergence of reciprocal exchange in animals and should be accepted with minimal extra exploration and a few additions and changes ot the text.

Reviewer 3 ·

Basic reporting

This is a paper that I found quite difficult to evaluate. I did not find any particular problem (though I should add that I am not a theoretician and hence cannot evaluate the mathematics/programming of the model). However, the question addressed is peculiar. I accept that Brosnan & de Waal are important figures in the field but for an evolutionarily minded reader it is pretty obvious that symmetry-based reciprocity cannot work in the long run. I do not even see why such a behavioural rule should be labelled reciprocity in the first place. There is nothing contingent on partner behavior. As far as I am aware, nobody has yet called altruism a form of reciprocity. As the authors point out there are similar ‘tag models’ for greenbeard that already conclude that tag strategies are unlikely to persist even though for greenbeard the underlying assumptions make more sense. Thus, the current model shows the obvious (though if the authors are right for the first time). If anything, one could have asked what sort of cognitive constraints or population structures might make such decision rules viable but this was not done. I see some parallels between symmetry based reciprocity and generalized reciprocity: simple but not obvious. I don’t whether the authors would want to discuss this.

Experimental design

does not apply

Validity of the findings

see comments below

Additional comments

Referee comments.

This is a paper that I found quite difficult to evaluate. I did not find any particular problem (though I should add that I am not a theoretician and hence cannot evaluate the mathematics/programming of the model). However, the question addressed is peculiar. I accept that Brosnan & de Waal are important figures in the field but for an evolutionarily minded reader it is pretty obvious that symmetry-based reciprocity cannot work in the long run. I do not even see why such a behavioural rule should be labelled reciprocity in the first place. There is nothing contingent on partner behavior. As far as I am aware, nobody has yet called altruism a form of reciprocity. As the authors point out there are similar ‘tag models’ for greenbeard that already conclude that tag strategies are unlikely to persist even though for greenbeard the underlying assumptions make more sense. Thus, the current model shows the obvious (though if the authors are right for the first time). If anything, one could have asked what sort of cognitive constraints or population structures might make such decision rules viable but this was not done. I see some parallels between symmetry based reciprocity and generalized reciprocity: simple but not obvious. I don’t whether the authors would want to discuss this.

Some specific points
The authors should define reciprocity early on. This is important as the symmetry based reciprocity certainly does not fit any definition I am aware of.
Partner choice versus partner control: on a theoretical level the distinction does not make sense. Noë and colleagues contrasted the two because they wanted to shift the attention to payoff distributions. However, avoiding cheaters is just another control mechanism. Also, calling a tag model a ‘partner choice model’ certainly should not include reference to Noë because his ideas of a market rely on choosing the partner that offers the highest payoff and nothing else. There is no market in the current paper, no supply and demand, no bargaining. The authors should rather emphasise the difference between their way of thinking and the biological market idea in order to avoid confusion.
For the models: it would be nice to see explicit life cycles with the important steps in a kind of bullet point format to get a quick grasp of each model.
The similarity between the model and real data in the first part is obviously purely phenomenological. Therefore, I am not sure that the first model is needed at all. Kin structure will already lead to preferred partners in the real data. Given the mismatch, I certainly would not call the first results ‘important’ as the authors do.
It would be good to see references to empirical papers where aiuthors concluded that they provide evidence for this weird concept. If the only examples are that authors fail to find contingent reciprocity and hence mention in the discussion something like ‘by the way Brosnan and de Waal proposed symmetry-based reciprocity so maybe there is something like this happening’ then the concept never took off at al.
Finally, regarding the literature on reciprocity: it is extremely incomplete. So many people thought about it and all the authors do is to refer to their own concept of emotional reciprocity, a concept that is almost impossible to demonstrate.

---

## Round 0.2 · accepted · Accept

Two reviewers have examined your revision and have concluded that you addressed the concerns adequately. I agree with them.

Reviewer 1 ·

Basic reporting

The intent and explanation of the work is now excellent. The review of the literature is good, grounding the manuscript.

A few minor typos:

Fix the sentence starting in line 209: In order to keep population size stable, the 20% of agents with the highest fitness was made replicate themselves

Line 320: “constraints” should be “constrains”

Line 329: *Symmetry-based reciprocity has the opposite shortcoming.

Experimental design

The experimental design is well defined and carried out with transparency. The clarifications on the intent of the non-evolutionary experiment help articulate the author's intent. Further, the newest draft does well to place their work in the large literature surrounding prosocial behaviour.

Validity of the findings

The findings are clear, whilst symmetry based reciprocity generates behaviour which appears cooperative, the behaviour is not evolutionarily viable. Consequently, it should be dismissed as a potential proximate mechanism for prosociality. Further, the discussion of the findings is balanced, and acknowledges the experiment's limitations.

Potentially a small point, however, I notice that the initial strategy distributions in the evolutionary model are 45/5, 25/25, and 5/45. Since, as the authors acknowledge, an agent-based model cannot validate all potential population frequency distributions, I'm curious why they didn't take the limit case? Wouldn't the validity of the experiments be increased if the initial strategy distributions were run at 50/0, 25/25, and 0/50? If defectors were only permitted to mutate into the population, then it would be a limit case test for invasion by scarcity.

In this particularly instance, I doubt it will alter the results. However, it is possible that including a population of 5% defectors is different than 0%. This is the case in the indirect reciprocity literature where, in some cases, cooperation is only sustained if the initial population of defectors is quite small. Again, I doubt it matters in this case, but it might be nice for completion sake and limit case testing.

Additional comments

This work tests whether symmetry-based reciprocity is, theoretically, a viable proximate mechanism for cooperation. The authors demonstrate that, whilst symmetry-based reciprocity generates behaviour we'd expect to see from a proximate mechanism, the strategy is not viable over an evolutionary time-frame. As such, the authors (barring an analytical proof which asserts otherwise) reject the notion that symmetry-based reciprocity could have evolved as a proximate mechanism for prosocial behaviour.

In general, this work explains and analyses the question well and warrants publication.

Reviewer 2 ·

Basic reporting

no further comments

Experimental design

no further comments

Validity of the findings

no further comments

Additional comments

I congratulate the authors on their satisfactory revision of the manuscript.
I have no further comments.